

# scAnnoX: an R package integrating multiple public tools for single-cell annotation

Xiaoqian Huang[1,*], Ruiqi Liu[1,*], Shiwei Yang[1], Xiaozhou Chen[1] and Huamei Li[2]

[1] School of Mathematics and Computer Science, Yunnan Minzu University, Kunming, Yunnan Province, China
[2] Department of Hepatobiliary Surgery, the Affiliated Drum Tower Hospital, Medical School, Nanjing University, Nanjing, Jiangsu Province, China
* These authors contributed equally to this work.

## ABSTRACT

**Background:** Single-cell annotation plays a crucial role in the analysis of single-cell genomics data. Despite the existence of numerous single-cell annotation algorithms, a comprehensive tool for integrating and comparing these algorithms is also lacking.

**Methods:** This study meticulously investigated a plethora of widely adopted single-cell annotation algorithms. Ten single-cell annotation algorithms were selected based on the classification of either reference dataset-dependent or marker gene-dependent approaches. These algorithms included SingleR, Seurat, sciBet, scmap, CHETAH, scSorter, sc.type, cellID, scCATCH, and SCINA. Building upon these algorithms, we developed an R package named scAnnoX for the integration and comparative analysis of single-cell annotation algorithms.

**Results:** The development of the scAnnoX software package provides a cohesive framework for annotating cells in scRNA-seq data, enabling researchers to more efficiently perform comparative analyses among the cell type annotations contained in scRNA-seq datasets. The integrated environment of scAnnoX streamlines the testing, evaluation, and comparison processes among various algorithms. Among the ten annotation tools evaluated, SingleR, Seurat, sciBet, and scSorter emerged as top-performing algorithms in terms of prediction accuracy, with SingleR and sciBet demonstrating particularly superior performance, offering guidance for users. Interested parties can access the scAnnoX package at https://github.com/XQ-hub/scAnnoX.

## INTRODUCTION

Single-cell sequencing data provide a high-resolution gene expression perspective within individual cells (*Wen et al., 2023*), revealing functional and phenotypic differences among individual cells (*Balzer et al., 2021*; *Kolodziejczyk et al., 2015*; *Rossin, Sobrin & Kim, 2021*; *Slovin et al., 2021*) and thereby revealing the diversity and heterogeneity within cell populations (*Bod et al., 2023*; *Chen et al., 2023*; *Fu et al., 2021*; *Hickey et al., 2023*; *Wang et al., 2022*). However, when analyzing single-cell data, identifying cell identities is essential

Corresponding authors
Xiaozhou Chen,
ch_xiaozhou@163.com
Huamei Li, li_hua_mei@163.com

and particularly critical (*Brendel et al., 2022*; *Cheng et al., 2023*; *Kim et al., 2021*). Currently, two main strategies are available for single-cell identity annotation: manual annotation and automatic annotation. Manual annotation consumes a significant amount of time, while automatic annotation is more convenient and rapid than manual annotation and is currently trending (*Abdelaal et al., 2019*; *Huang & Zhang, 2021*).

As researchers have increasingly focused on single-cell identity annotation tasks, numerous automatic annotation algorithms have emerged. One type of method involves auto annotation based on the marker genes associated with cell types and scoring the presence of these marker genes in cell clusters (*Pasquini et al., 2021*), referred as marker-based annotation. Examples illustrating this principle include algorithms such as scCATCH (*Shao et al., 2020*), sc.type (*Ianevski, Giri & Aittokallio, 2022*) and SCINA (*Zhang et al., 2019*). The second type of method requires a reference dataset containing cell type information for calculating the similarity between the expression profiles of query genes and the reference dataset and referred reference-based annotation. Prominent examples of this style of method include SingleR (*Aran et al., 2019*) and sciBet (*Li et al., 2020*).

Generally, annotation algorithms exhibit unique applicability and constraints. Reference-based tools like SingleR and scmap (*Kiselev, Yiu & Hemberg, 2018*) rely on statistical metrics, while CHETAH (*de Kanter et al., 2019*) employs a hierarchical classification approach. Marker-based tools like cellID (*Cortal et al., 2021*) leverage multivariate statistical methods, whereas SCINA utilizes bimodal distribution fitting for marker genes, and scCATCH relies on cellular heterogeneity within clusters. However, the autonomy of each annotation result presents difficulties in effectively studying, comparing, filtering, and optimizing the combined annotation results from different methods. Consequently, selecting the most appropriate algorithm for specific research objectives entails preprocessing data to adhere to the algorithm's specifications, conducting model tuning, and gaining a thorough understanding of the intricacies associated with each algorithm. This decision-making process often necessitates substantial time and effort investments.

In this context, the present study developed an R package known as scAnnoX, which amalgamates 10 distinct single-cell sequencing data-based cell identity recognition algorithms into a unified framework, facilitating a comparative analysis. The overarching goal is to assist researchers in efficiently analyzing scRNA-seq data, offering targeted guidance for making judicious decisions regarding the intricate selection of single-cell identity recognition algorithms and simplifying the processes of testing, evaluating, and comparing various algorithms within an integrated environment.

## MATERIALS AND METHODS

This research endeavored to construct an R package, denoted scAnnoX, designed to comprehensively amalgamate ten distinct algorithms for single-cell RNA sequencing data-based cell identity recognition. These algorithms include SingleR (version 1.8.1) (*Aran et al., 2019*), Seurat (version 4.3.0.1) (*Hao et al., 2023*), sciBet (version 0.1.0) (*Li et al., 2020*), scmap (version 1.9.3) (*Kiselev, Yiu & Hemberg, 2018*), CHETAH (version

1.14.0) (*de Kanter et al., 2019*), scSorter (version 0.0.2) (*Guo & Li, 2021*), sc.type (version NA) (*Ianevski, Giri & Aittokallio, 2022*), cellID (version 1.2.1) (*Cortal et al., 2021*), scCATCH (version 3.2.2) (*Shao et al., 2020*) and SCINA (version 1.2.0) (*Zhang et al., 2019*). In each instance, source code packages were diligently installed, or scripts were meticulously sourced from GitHub repositories. Evaluating the performance of ten single-cell identity recognition algorithms is a multifaceted endeavor that necessitates the establishment of clearly defined methodologies and the implementation of a rigorous set of standardized experimental procedures.

## Data preprocessing

In the initial stage, the primary tasks for users involved acquiring the processing single-cell sequencing data into Seurat objects. If the data had not been initialized, normalization could be performed using the "NormalizeData" function with the "LogNormalize" parameter (accessible through the "Seurat" package). Alternatively, principal component analysis (PCA)-based (*Lever, Krzywinski & Altman, 2017*) dimensionality reduction could be applied to the data using the "RunPCA" function. Data preprocessing could also be carried out using the "commonClustering" function of the scAnnoX package. This process facilitated a deeper understanding of the similarities and differences between individual cells.

## Reference profile acquisition and marker gene selection

When invoking tools based on reference datasets, users are required to provide corresponding reference expression profiles based on specific research organizations or particular cell types. The use of marker gene-based tools enables the identification of marker genes from the reference expression profiles. This can be achieved using the "FindAllMarkers" function within the "Seurat" package. It is worth noting that if the users have already provided a list of marker genes, the marker gene identification step is not executed.

## Algorithm selection

After conducting an in-depth exploration of various single-cell annotation algorithms, we selected ten widely used and publicly available algorithms based on their different methods and functionalities (Table 1). These algorithms can be categorized into two main classes. One class of tools, including scSorter, sc.type, cellID, scCATCH, and SCINA, relies on marker genes associated with specific cell types. The other class of tools, including SingleR, Seurat, sciBet, scmap, and CHETAH, utilizes information from reference cell type datasets.

## Integration of annotations

This task involved integrating and optimizing ten different single-cell RNA sequencing data-based cell identity recognition algorithms. Each of these algorithms has unique strengths and applications; therefore, cleverly combining them will provide researchers with a broader range of choices and more powerful tools. This effort aimed to enhance the diversity of data processing, thereby improving the feasibility of research. To optimize the algorithm integration process, we needed to delve into the performance and characteristics

**Table 1 Publicly available annotation tools for single cells.**

| AnnoTool | Author | Version | Method | Depend | Reference or marker-based | Publication |
|---|---|---|---|---|---|---|
| SingleR | Aran D et al. | 1.8.1 | Correlation to training set | R | RB | 2019 |
| Seurat | Hao Y et al. | 4.3.0.1 | Built-in functions | R(>= 4.0.0) | RB | 2023 |
| sciBet | Li C et al. | 0.1.0 | Multinomial model | R | RB | 2020 |
| scmap | Kiselev VY et al. | 1.9.3 | Nearest median classifier | R(>= 3.4) | RB | 2018 |
| CHETAH | de Kanter JK et al. | 1.14.0 | Correlation to training set | R(>= 4.0) | RB | 2019 |
| scSorter | Guo H et al. | 0.0.2 | Marker genes | R(>= 3.6.0) | MB | 2021 |
| sc.type | Ianevski A et al. | NA | Specificity of marker genes | R | MB | 2022 |
| cellID | Cortal A et al. | 1.2.1 | Multivariate statistical methods | R(>= 4.1) | MB | 2021 |
| scCATCH | Shao X et al. | 3.2.2 | Cell heterogeneity in the clusters | R(>= 4.0.0) | MB | 2020 |
| SCINA | Zhang Z et al. | 1.2.0 | Bimodal distribution fitting for marker genes | R(>= 2.15.0) | MB | 2019 |

of these different algorithms and find the best way to integrate them to ensure that they could work together while considering the quality and characteristics of data.

The scAnnoX package includes a function called "autoAnnoResult", which was used to aggregate and summarize the predictions of the 10 algorithms. After aggregation, following the principle of the voting algorithm, the frequency (denoted as $N_{Cell\ type}$.) of each prediction outcome produced for the same sample was computed and expressed as a ratio to the total number of methods, thereby yielding the frequency of each prediction. The result of the "autoAnnoResult" function was the prediction with the highest frequency and was determined by the following formula:

$$argmax_{Cell\ type}\left(p = \frac{N_{Cell\ type}}{N_{Tools}}\right).$$

This result serves as the final prediction in the scAnnoX package, which effectively integrates multiple algorithms. With this approach, researchers can analyze and interpret single-cell RNA sequencing data, providing them with more powerful tools and a wider range of choices for more effectively conducting scientific research.

### Experimental validation

Datasets originating from diverse organizational sources and various data platforms were partitioned into test and reference sets at a 6:4 ratio. The test set was used for an algorithmic performance evaluation, while the reference set was employed for model training or served as a performance benchmark. Leveraging the scAnnoX package, we conducted data annotation and validation, leveraging a suite of functions for assessing the precision and consistency of single-cell RNA sequencing data. We scrutinized the alignment between the predictions produced by each method on the test set and the ground-truth labels within the reference set. Two different performance metrics, the prediction accuracy and the root mean square error (RMSE), were used to measure the accuracy and reliability of each algorithm, which facilitated the selection of the most

appropriate algorithm for satisfying research needs. This approach helped to determine which algorithms performed well in terms of validating multisource data.

## Performance assessment

### *Accuracy*

Performance metrics serve as measurement standards for appraising the efficacy of models, algorithms, or systems within the context of specific tasks. In this context, we present a pivotal performance metric: accuracy. Accuracy is a ubiquitous metric that is utilized to assess the effectiveness of classification models or algorithms. This metric gauge the ratio of samples correctly predicted by the model to the number of overall samples. The formula for calculating accuracy is succinctly expressed as follows:

$$Accuracy = \frac{N_{pred=Publiction}}{N}$$

where $N_{pred=Publiction}$ signifies the number of samples for which the predictions of the utilized model or algorithm align with the authentic labels, and $N$ denotes the aggregate number of samples.

### *Root mean square error of prediction performance*

The root mean square error is a statistical metric that quantifies the disparity between predicted and actual values (*Kim et al., 2021*). It is calculated as the square root of the mean of the squared differences between the predicted and actual values divided by the total number of observations. The RMSE is particularly sensitive to atypical data points, which are often referred to as outliers, making it a valuable tool for assessing the overall accuracy and robustness of predictive models. The formula for the RMSE is defined as follows:

$$RMSE = \sqrt{\frac{1}{N}\sum_{i=1}^{n}(true_i - pred_i)^2}.$$

In this equation, $N$ denotes the total number of experiments conducted, $n$ signifies the number of predicted samples, $true_i$ represents the true value for the sample, and $pred_i$ is indicative of the predicted value for the same sample.

## RESULTS

## R package development for single-cell RNA sequencing data annotation

Utilizing the R programming language, we have successfully engineered an R package called scAnnoX. This meticulously crafted package integrates a comprehensive suite of 10 distinct annotation algorithms, as previously elucidated. Each of these algorithms exhibits unique applicability and inherent limitations. To help users conduct their research more efficiently using the scAnnoX package, we meticulously crafted a comprehensive user guide for scAnnoX. These user-friendly instructions ensure the effortless accessibility and operation of all the integrated annotation algorithms.

The scAnnoX software package employs distinct strategies based on algorithmic classification. For algorithms relying on a reference dataset, both a reference dataset and a test dataset are required as inputs. Conversely, for algorithms centered on marker genes, a reference dataset and marker genes serve as the inputs. The ultimate output of the scAnnoX software package integrates the annotation results acquired from both types of algorithmic approaches. Furthermore, we meticulously fine-tuned and optimized this R package, ensuring not only its stability but also its efficiency, thus guaranteeing its long-term maintainability and scalability. The specific architecture of the scAnnoX package is illustrated in Fig. 1.

## Use of the scAnnoX package

The input data for the scAnnoX software package must adhere to the requirements of the Seurat data format, ensuring that the columns of the dataset include gene expression values and cell identity information. Subsequently, preprocessing steps, such as normalization and dimensionality reduction, are applied to the data. If necessary, annotated algorithmic functions can be invoked using the "listToolMethods" function of the R package. The test dataset is subsequently annotated using the "autoAnnoTools" function provided in the package. "autoAnnoTools" relies primarily on two fundamental parameters, "method" and "strategy", with optional parameters, including the reference dataset, reference cell types, and marker gene information, defaulted to NULL. The "method" parameter refers to the name of the single-cell annotation tool, while the "strategy" parameter indicates the categorization of the single-cell annotation tool (see the Materials and Methods section for details).

The necessity of these optional parameters depends on the value of the underlying method. If the method is a marker-based algorithm, marker gene information must be provided. If the method is a reference-based approach, both the reference dataset and reference cell types need to be provided.

The output of the "autoAnnoTools" function contains the annotation results produced by the chosen single-cell identity recognition algorithm for the samples within the testing dataset. These annotation results assist in determining the single-cell identity of each sample. Usage examples of the scAnnoX package can be found at https://github.com/XQ-hub/scAnnoX/tree/main/vignettes. R, with the code and output results provided therein. These examples serve as a valuable reference for understanding the practical implementation of the package employed in your research.

## Annotation for assessing the accuracy of internal datasets

To substantiate and compare the precision levels of the ten annotation tools, datasets emanating from diverse tissue origins and distinct data acquisition platforms were partitioned into experimental test sets and reference sets, maintaining a 6:4 ratio. Comprehensive scrutiny was utilized to rigorously assess the efficacy of these computational algorithms within the confines of the given datasets. In this study, we conducted a comprehensive algorithmic performance evaluation using the scAnnoX software package on four distinct single-cell omics datasets. Specifically, our analysis

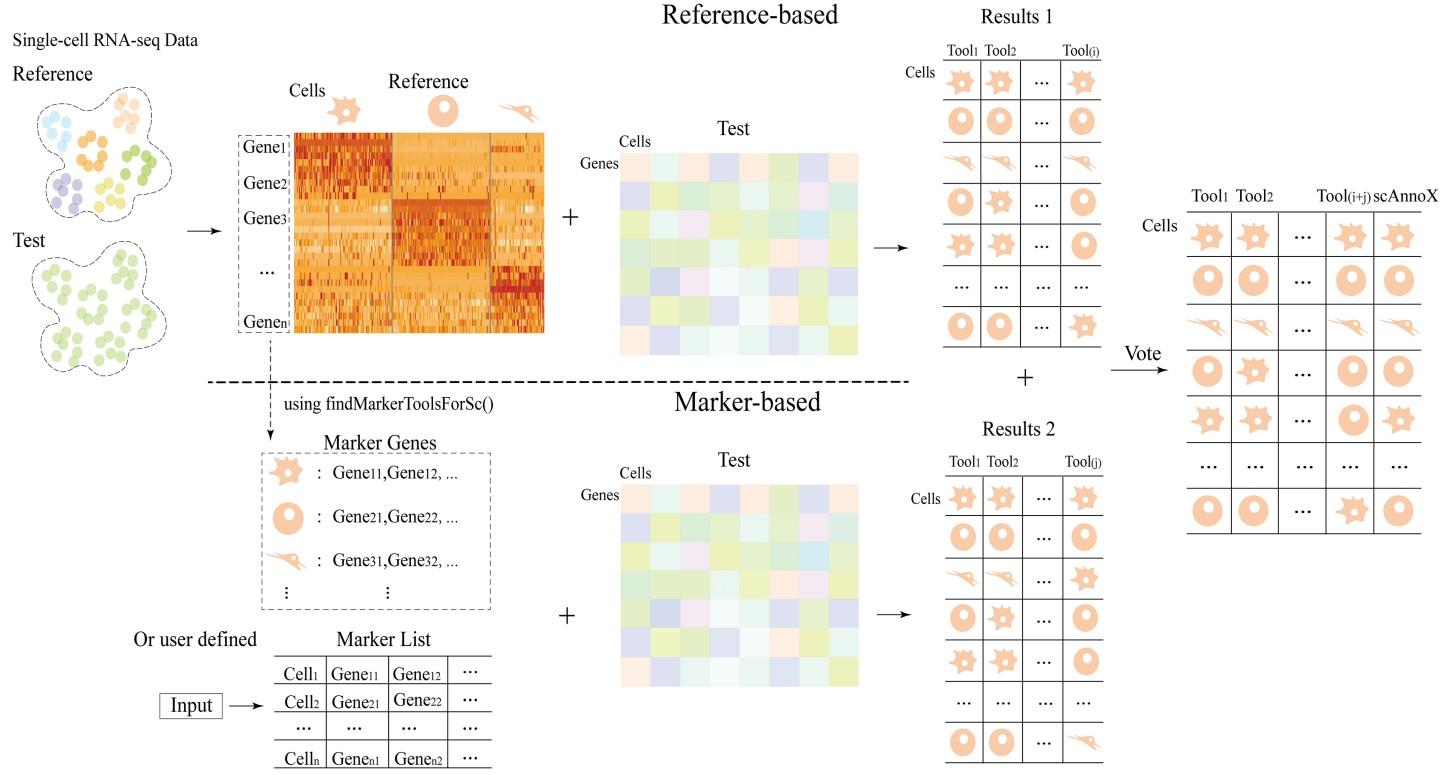

**Figure 1 The integration process architecture used for the scAnnoX package.**

centered on the human islet cell dataset published by *Xin et al. (2016)*, where the scSorter and SCINA algorithms exhibited exceptional capabilities, achieving an outstanding classification accuracy of 99.69% (Fig. 2A). Furthermore, we extended our assessment to include the human liver tissue dataset presented by *Camp et al. (2017)* and the human brain transcriptome dataset developed by *Darmanis et al. (2015)*, where the sciBet algorithm demonstrated remarkable performance, with classification accuracies of 98.43% and 87.83%, respectively (Figs. 2B, 2C). In the case involving the human liver tissue dataset, the sc.type algorithm also achieved a classification accuracy comparable to that of sciBet. Of particular significance was the performance of the SingleR algorithm, which achieved an impressive accuracy of 88.89% in terms of classifying the cell types within the human brain transcriptome dataset and an exceptional accuracy of 96.17% on the adult mouse cortical cell dataset produced by *Tasic et al. (2016)* (Fig. 2D). In contrast, the performance of the cellID method was comparatively inferior, demonstrating an accuracy of 61.78% on the human liver tissue dataset and a mere 12.91% accuracy on the human pancreatic islet cell tissue dataset. Furthermore, it is noteworthy that the performances of scmap and scCATCH, while competitive in certain contexts, exhibited considerable variability and susceptibility to the characteristics of diverse datasets.

Based on these evaluations, we utilized the integrated results obtained through the built-in functionalities of the autoAnnoTools function within the scAnnoX software package. As exemplified by the human islet cell and human liver tissue datasets, we produced two-dimensional visualizations of the original cell types, the scAnnoX

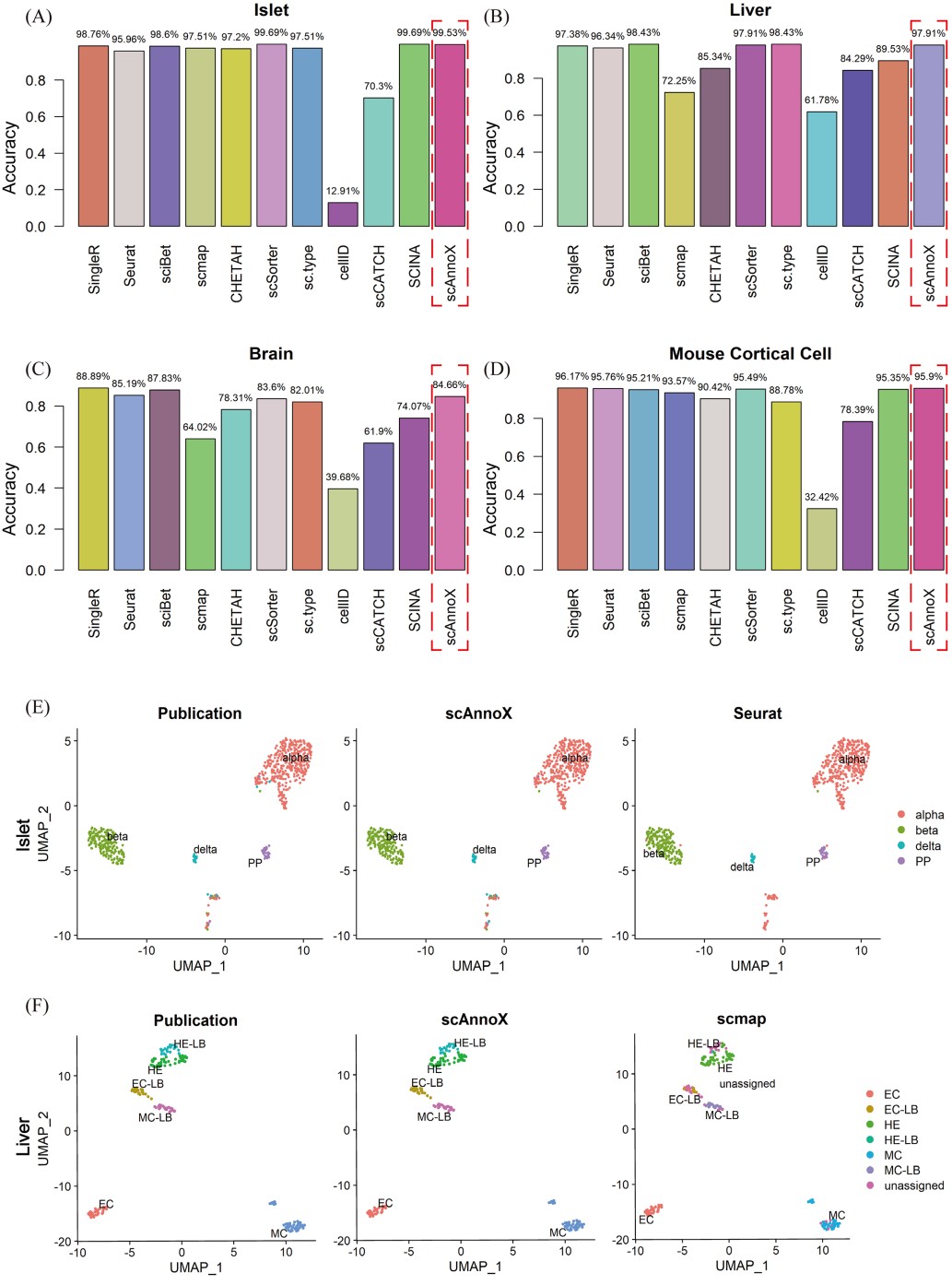

**Figure 2 Performance evaluation outcomes of ten algorithms and integrated results obtained on four independent datasets.** (A) Accuracy assessment results obtained on the pancreatic tissue dataset. (B) Accuracy assessment results obtained on the liver tissue dataset. (C) Accuracy assessment results obtained on the brain tissue dataset. (D) Accuracy assessment results obtained on the mouse cortical cell tissue dataset. (E) Cellular type visualization results obtained on the pancreatic tissue dataset: the real cell types, scAnnoX-predicted results, and Seurat-predicted results. (F) Cellular type visualization results obtained on the liver tissue dataset: the real cell types, scAnnoX-predicted results, and scmap-predicted results.
package-predicted cell types, and those predicted by one of the algorithms (Figs. 2E, 2F). The uniform manifold approximation and projection (UMAP) visualization approach demonstrated remarkable stability and robust performance within the integrated results.

## Precision assessment of cross-platform datasets

The diversity of scRNA-seq techniques offers a valuable opportunity for validating cross-platform datasets derived from the same biological tissue. To substantiate this assertion, we conducted a precision assessment experiment on cross-platform datasets using two independent and well-sequenced datasets originating from different sequencing platforms. The primary objective of this study was to comprehensively and systematically evaluate the performance of the scAnnoX package. Two distinct datasets were subjected to validations in this experiment: one sourced from pancreatic tissue, as reported by *Xin et al. (2016)* and *Lawlor et al. (2017)*, and another derived from thymic tissue, as reported by *Yasumizu et al. (2022)* and *Park et al. (2020)*. Random subsampling was applied to each dataset set obtained from the different platforms; one dataset was used as the reference dataset, and the other was used as the test dataset. We compared the annotation accuracies of the ten annotation algorithms embedded within the scAnnoX package and subsequently derived an integrated annotation accuracy metric.

In the context of the cross-platform pancreatic tissue dataset, we employed the dataset curated by *Xin et al. (2016)* as a reference training set and utilized the dataset curated by *Lawlor et al. (2017)* as a test set. Our primary focus was on assessing the predictive performance of various algorithms and the use of the scAnnoX software package for identifying shared cell types between the two datasets (Figs. 3A, 3B). The results of this validation exercise unequivocally highlight the robustness of the majority of the examined tools in terms of accurately characterizing and annotating the test dataset. Specifically, SingleR, sciBet, scSorter, and SCINA exhibited remarkable predictive accuracies of 99%, while Seurat achieved an accuracy of 96.59%. Notably, cellID misclassified a significant number of beta cells as alpha cells (Fig. 3A). SingleR demonstrated suboptimal performance with respect to identifying pancreatic polypeptide-secreting cells (PPs) and delta cell types, whereas Seurat exhibited deficiencies in terms of recognizing delta cell types (Fig. 3C). A closer examination revealed that the challenges in distinguishing these cell types could be attributed to their relatively low cell counts, especially the scarcity of pancreatic polypeptide-secreting cells within the islet dataset (Fig. 3A). Remarkably, scAnnoX, which leverages integrated annotations, was the top-performing approach, with an impressive accuracy of 99.69%. This exceptional performance was particularly striking because of its perfect prediction accuracy of 100% for alpha, beta, and delta cell types, surpassing the performance of the other algorithms (Fig. 3C).

In the context of a multiplatform thymic tissue dataset, we assessed the reference dataset produced by *Park et al. (2020)* using it as the baseline for a validation against the dataset provided by *Yasumizu et al. (2022)*. Given the high cell types heterogeneity and the limited sample sizes within certain cell type categories, we comprehensively reclassified and aggregated the cell types within the dataset. Specifically, we combined subtypes such as mTEC(I), mTEC(II), mTEC(III), and mTEC(IV) into a unified category referred to as
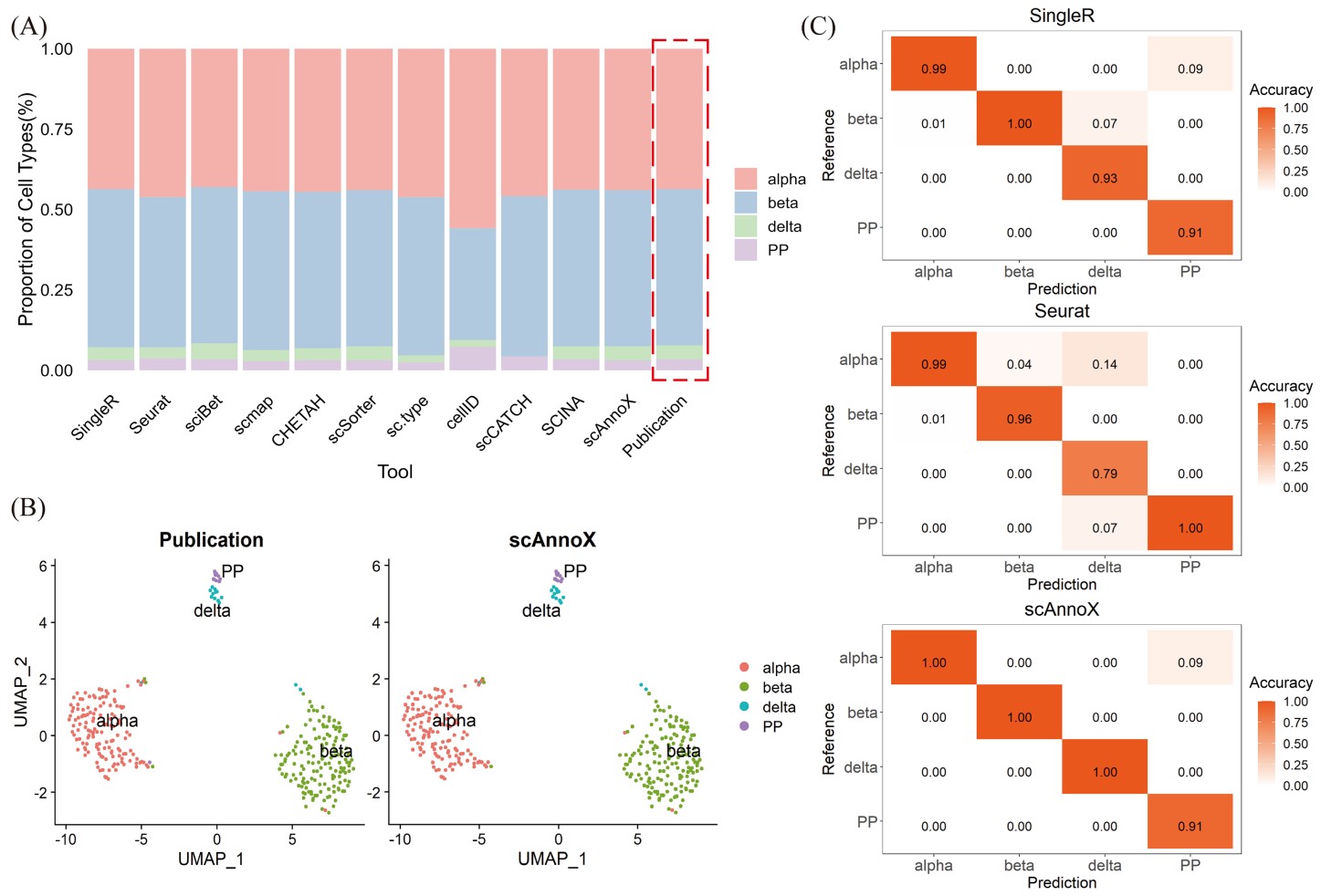

**Figure 3** **Performance evaluation results obtained on independent cross-platform islet datasets.** (A) Proportions of the four cell types in the test dataset. (B) UMAP visualization of the cell types contained in the dataset of *Lawlor et al. (2017)* and the integrated cell types derived from the scAnnoX software package. Each cluster represents a distinct cell type. (C) Prediction accuracy of SingleR, Seurat, and scAnnoX for the four cell types.

"mTEC", while consolidating subtypes such as DC1, DC2, and aDC into a category denoted as "DC". Subsequently, we determined the predictive accuracy achieved for each cell type (Fig. 4A). Following the data preprocessing steps, we proceeded to evaluate the annotation performance of various computational algorithms. Due to the intricate nature of the cell types and the potential confounding effects of batch processing, the validation results were not satisfactory. The accuracies of most intrinsic methods tended to converge within the range of 45% to 66% (Fig. 4B). Notably, after the integration process, scAnnoX achieved an accuracy of 67.2%. However, it is worth mentioning that the misclassification of cell types was predominantly centered on the fine-grained subtyping of B cells and T cells (Figs. 4C, 4D).

This investigation underscores the complexities inherent in single-cell omics data analysis tasks, particularly in the context of intricate cell type distinctions, and it highlights

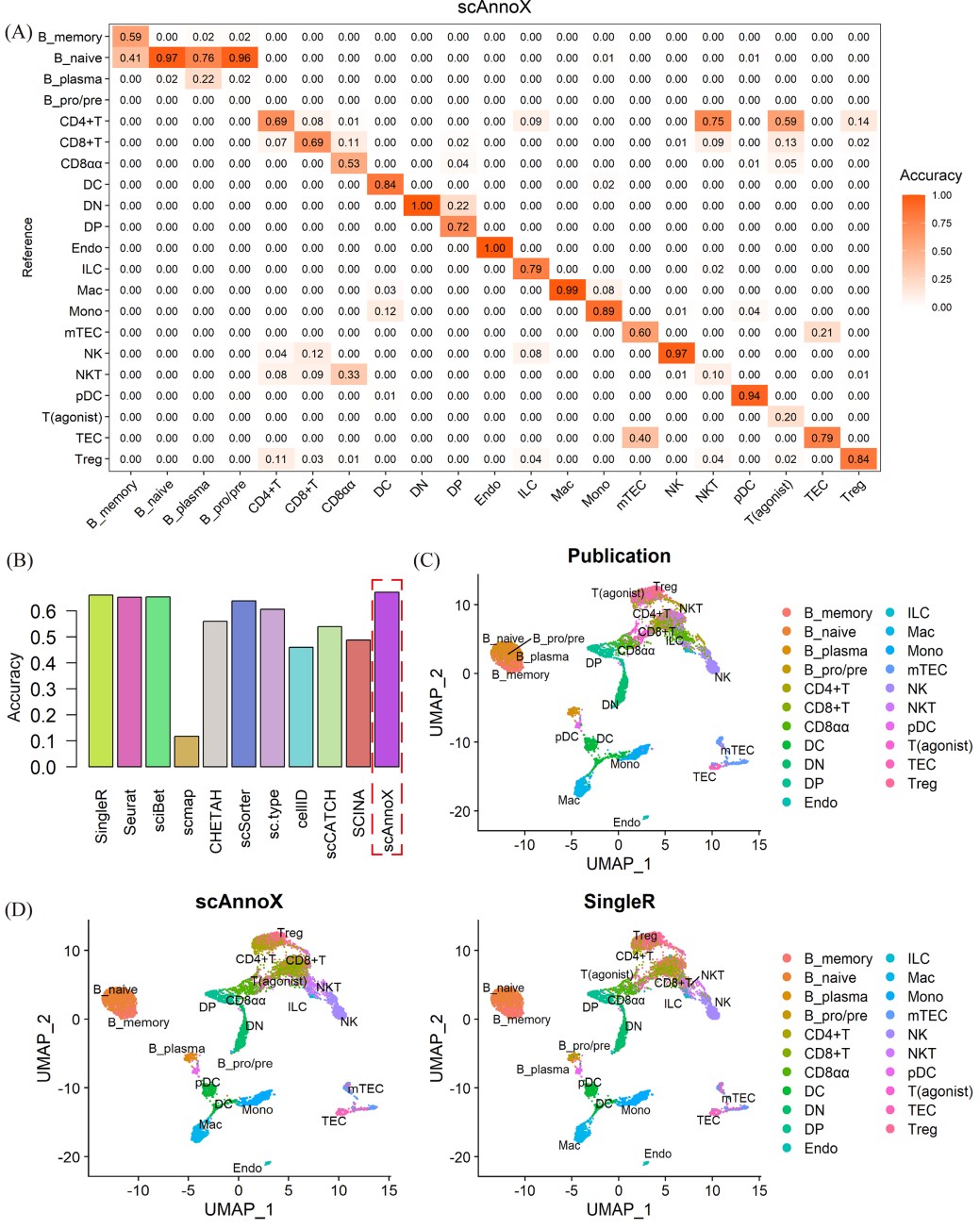

**Figure 4 Performance assessment results obtained on cross-platform thymic datasets.** (A) Evaluation of the predictive performance achieved for every cell type in the test dataset using the integrated results of the scAnnoX package. (B) Comparative assessment of the predictive performance of ten algorithms and the integrated results of the scAnnoX attained on the dataset of *Yasumizu et al. (2022)*. (C) The original cellular subtype distribution based on the dataset published by *Yasumizu et al. (2022)* Each cluster represents distinct cellular subtypes. (D) UMAP visualization of the cell types predicted by scAnnoX and SingleR.

the importance of algorithmic enhancements for increasing the accuracy of cell type annotations.

## Stability assessment of the annotations produced by scAnnoX

In the context of these experiments, we have successfully achieved a high predictive accuracy for the integrated results. Furthermore, we conducted a comprehensive analysis to assess the robustness and reliability of these integrated findings.

We comprehensively integrated and synthesized of all the experiments, generating integrated predictions for each scenario. By leveraging the built-in functionalities of the scAnnoX software package, we obtained integration results for each experiment. In comparison with various algorithms, scAnnoX consistently exhibited outstanding predictive performance, maintaining a consistently high level of accuracy, as depicted in Figs. 5A and 5B. To further assess the robustness and flexibility of the scAnnoX-integrated results, we calculated the root mean square error between the prediction results and the actual cell types (Fig. 5C). This evaluation unequivocally demonstrated the stability and resilience of the integration results provided by the scAnnoX software package. Additionally, our study underscores the significant capability of integration results to mitigate the adverse effects of data sparsity and batch effects relative to individual algorithms. This enhanced robustness and the exceptional observed performance further underscore the stability and reliability of our approach.

## Comparative computational runtime analysis

Building upon the experiments validating our in-house dataset, our study undertook a comprehensive analysis that unveiled profound disparities among the 10 distinct single-cell identity recognition algorithms concerning their computational execution times. This investigation underscores the significance of our work by shedding light on the temporal dynamics of these algorithms, which is a crucial dimension in the ever-evolving landscape of single-cell omics research.

In the pancreatic islet cell dataset, we grappled with an extensive volume of data, encompassing 38,008 genes and 1,809 samples. Notably, the sc.type and the SCINA methods exhibited exceptional efficiency in this scenario, completing the analysis within 0.30 and 0.55 seconds, respectively (Fig. 6A). In fact, they boasted the shortest processing times among the ten algorithms we evaluated, an achievement that merits strong emphasis. Conversely, scSorter and cellID required longer durations to complete the task. On the liver and brain tissue datasets, which feature 465 and 466 samples, respectively, and approximately twenty thousand genes, sc.type and SCINA continued to exhibit outstanding performance, with execution times remaining under 0.6 s and even dipping to 0.3 s in the case of the hepatic tissue dataset (Figs. 6B, 6C). On the mouse cortical cell dataset, which included 1,600 and 1,809 samples, and harbored complex cell types, sc.type still managed to provide predictions within 0.5 s, while scCATCH required 221.75 s (Fig. 6D). Taken together, the results of these experiments indicate that sc.type and SCINA consistently exhibited highly favorable performance, whereas scSorter and cellID required

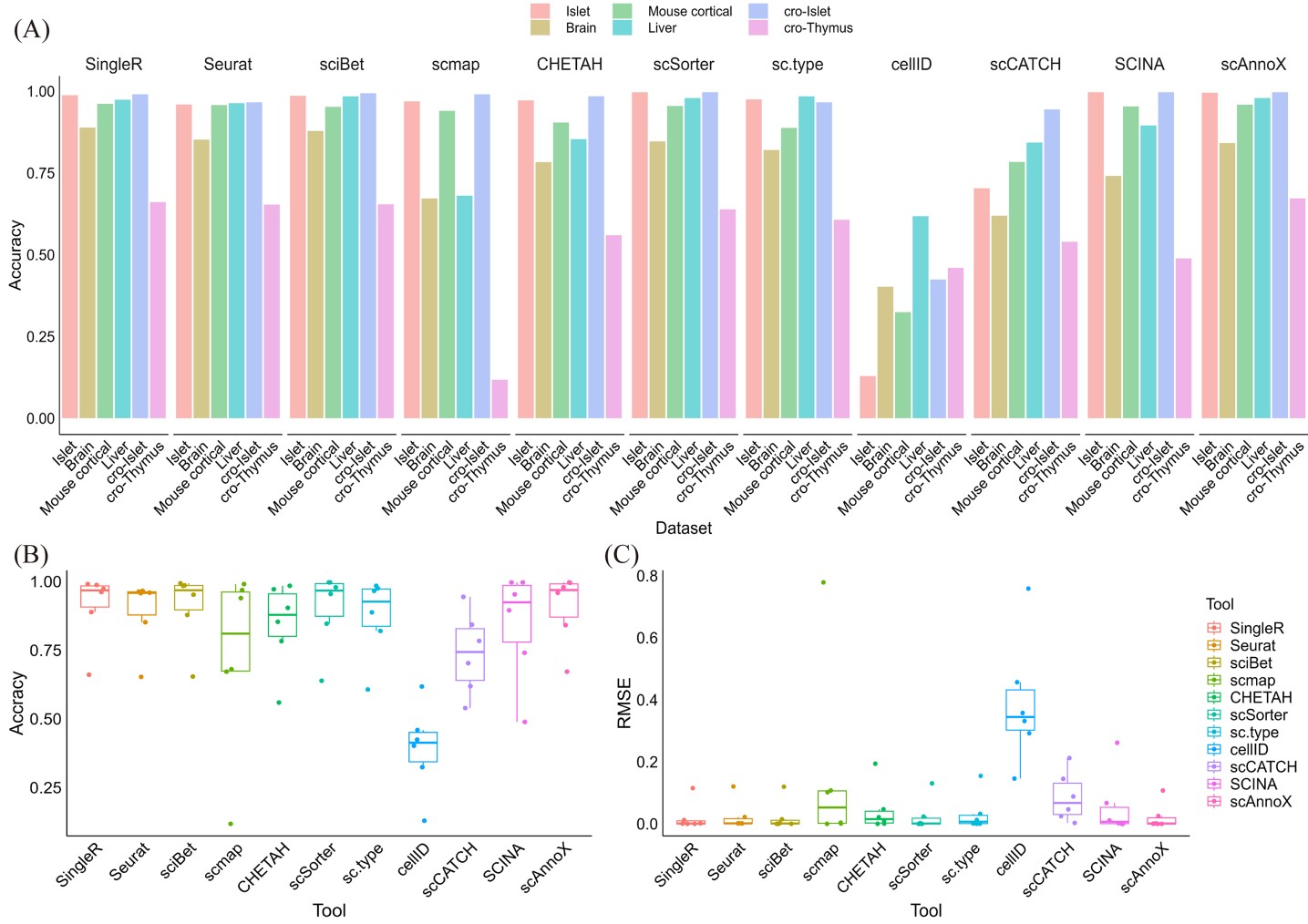

**Figure 5 Comprehensive comparative analysis of the predictive performance of different algorithms.** (A) Predictive performance comparison among all algorithms across all prediction experiments. (B) Evaluation of the prediction accuracy achieved for all algorithms and the integrated scAnnoX method across all experiments. (C) Assessment of the root mean square errors induced by all algorithms and the integrated scAnnoX method across all experiments.

longer durations to complete their tasks. scCATCH demonstrated an increase in runtime when faced with datasets featuring complex cell types.

The experimental analysis results regarding the running times of various algorithms across different datasets reveal a noteworthy trend: a substantial increase in the sample size or data complexity level corresponds to an increase in time consumption. To be more specific about the runtime implications, certain algorithms exhibited substantial variations, while others remained relatively stable.

Initially, our observations demonstrated significant increases in the running times of the CellID, sciBet, scmap, Seurat, and SingleR algorithms in response to enlarged sample sizes. This phenomenon is attributed to the necessity of processing an increased number of data points and performing more computationally demanding tasks. In contrast, the scCATCH algorithm displayed atypical behavior as the sample size increased. According to the

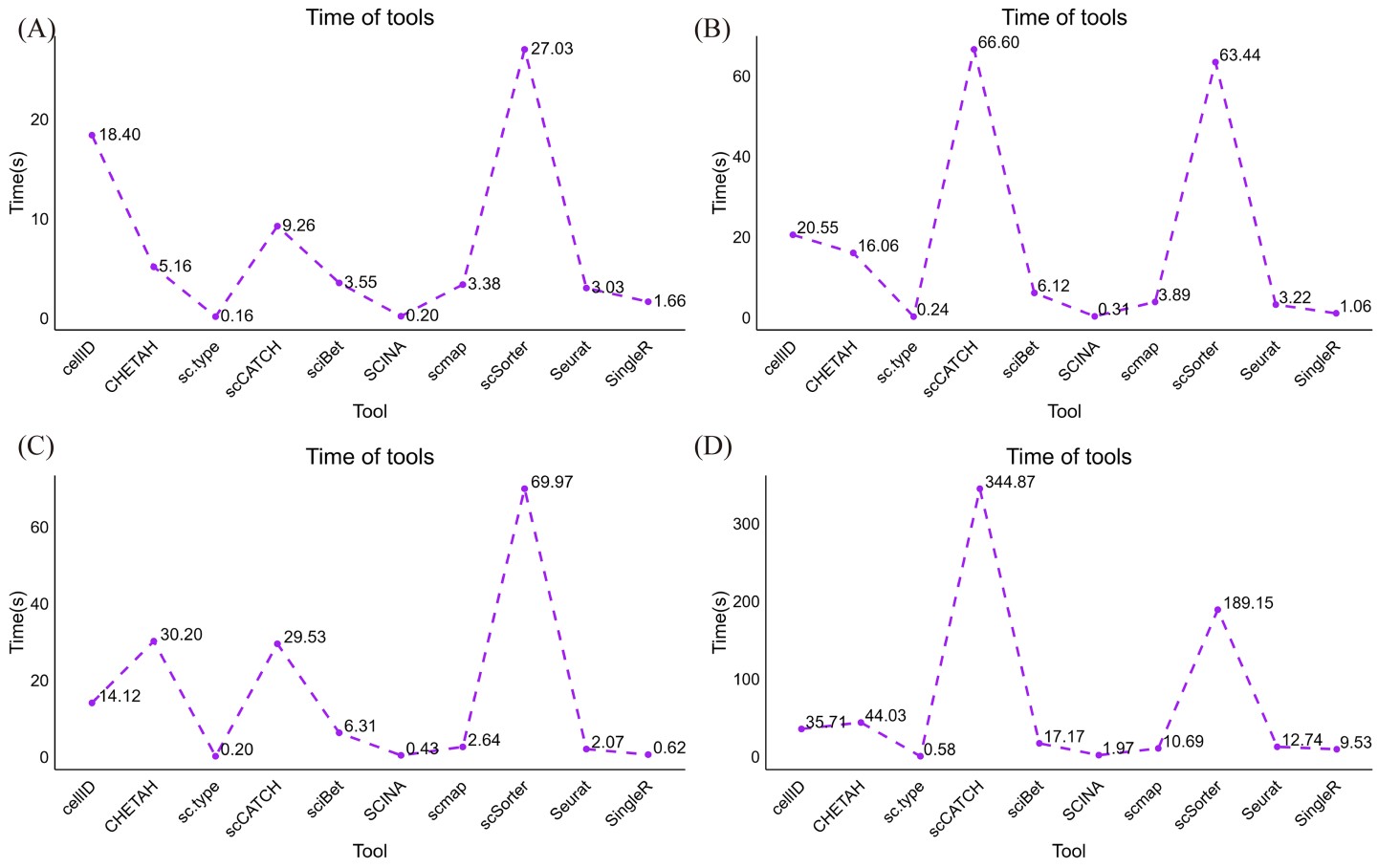

**Figure 6 Runtime distributions produced for datasets acquired from various organizational sources and platforms.** (A) Runtime distributions yielded by the 10 annotation tools on the human islet cell tissue dataset. (B) Runtime distributions produced by the 10 annotation tools on the human liver tissue dataset. (C) Runtime distribution yielded by the 10 annotation tools on the human brain transcriptome dataset. (D) Runtime distribution produced by the 10 annotation tools on the adult mouse cortical cell dataset.

comparative analysis between the human islet cell tissue dataset and the human liver tissue dataset, the running time of scCATCH decreased with increasing sample size. In contrast, certain algorithms, such as sc.type and SCINA, appear to have been less influenced by sample size variations. This observation underscores the superior stability and efficiency of these algorithms in handling extensive datasets.

In summary, these findings illuminate the varying performances of distinct algorithms when confronted with different sample sizes. Researchers should be mindful of the influence of their sample size when choosing an algorithm, ensuring that the selected algorithm aligns with their research requirements and can execute the analysis within a reasonable timeframe.

## DISCUSSION

With the advancement of single-cell RNA sequencing technologies, numerous single-cell annotation algorithms have emerged. Given the diverse requirements of different algorithms in terms of data preprocessing, input–output data formats, and other operations, researchers are faced with the daunting task of developing a profound

understanding of the algorithmic structures embedded in source codes, demanding substantial time and effort investments. In this context, our study established a comprehensive framework designed to accommodate ten prominent single-cell annotation algorithms. Within this framework, we developed an R package named "scAnnoX." This software package provides standardized data input and output architectures for these algorithms. Thus, researchers can streamline their workflows by engaging in a round of data preprocessing according to the input format specified by this package. This approach enhances the efficiency of the algorithm selection process and provides the single-cell genomics research community with a simplified and reproducible platform for testing and comparing various single-cell annotation tools.

Additionally, within scAnnoX, a function named "autoAnnoResult" was implemented. This function is utilized to generate comprehensive predictions in scAnnoX, and a subsequent validation conducted across various datasets demonstrated the commendable robustness of the performance achieved by scAnnoX. Researchers can leverage the scAnnoX package to flexibly select and validate one or multiple algorithms embedded within the package, facilitating comparative analyses among diverse algorithms. Researchers can also conduct extensive downstream analyses based on specific research objectives.

Specifically, we validated and compared the performances of ten algorithms, revealing temporal fluctuations in their performances across different datasets. Among these approaches, SingleR, Seurat, sciBet, and scSorter exhibited higher prediction accuracy, while SingleR, sciBet, sc.type and SCINA required the shortest annotation time, demonstrating superior efficiency. Notably, SingleR and sciBet ensured high prediction accuracy alongside greater efficiency. This empirical evidence aids in assessing the differences in performance among algorithms.

The development of scAnnoX has enabled effective analyses of single-cell RNA sequencing data, streamlining the processes of testing, evaluating, and comparing multiple algorithms, thereby equipping researchers with the tools necessary to navigate the complexity of algorithm selection. However, the annotation tools contained within scAnnoX are currently limited and represent one of the forthcoming research directions. Additionally, visualization modules tailored for single-cell RNA sequencing data can be incorporated into scAnnoX to better assist researchers in downstream analyses.

## CONCLUSIONS

Based on the field of single-cell transcriptomics, we developed an R package named scAnnoX by integrating ten different single-cell RNA sequencing data annotation algorithms. The development of the scAnnoX software package addresses the necessity of a universal input mode that is applicable to all single-cell RNA sequencing data identification algorithms, allowing for a deeper understanding of the intricacies of each algorithm. Consequently, researchers can utilize the scAnnoX package to obtain experimental results from these ten algorithms, further enhancing their prediction accuracy through the "autoAnnoResult" function. Thus, scAnnoX reduces the time and effort required for data preprocessing and model optimization. Employing the scAnnoX

software package, we conducted comparative evaluations of the predictive performance and execution times of the ten algorithms across different datasets, including internal validation experiments and cross-platform validation experiments. The results of the study underscore the critical role of scAnnoX in providing researchers with essential decision-making tools, enabling them to make informed choices based on their research objectives. These experimental findings not only validate the importance of comparing and integrating algorithms but also offer robust support for researchers who want to cautiously select algorithms in specific research scenarios. This approach provides a useful tool for analyzing single-cell RNA sequencing data and holds the potential to drive significant advancements in biomedical research.

## ACKNOWLEDGEMENTS

We thank all the authors involved in this study for data collection, preparation, quality control and manuscript writing. This work was carried out on the High-performance Computing platform of Yunnan Minzu University.

### Funding

This research was funded by the National Natural Science Foundation of China of XZC, grant number 31460297. The funders had no role in study design, data collection and analysis, decision to publish, or preparation of the manuscript.

### Grant Disclosures

The following grant information was disclosed by the authors:
National Natural Science Foundation of China of XZC, Grant number: 31460297.

### Competing Interests

The authors declare that they have no competing interests.

### Author Contributions

- Xiaoqian Huang conceived and designed the experiments, performed the experiments, analyzed the data, prepared figures and/or tables, authored or reviewed drafts of the article, and approved the final draft.
- Ruiqi Liu performed the experiments, analyzed the data, prepared figures and/or tables, and approved the final draft.
- Shiwei Yang analyzed the data, prepared figures and/or tables, and approved the final draft.
- Xiaozhou Chen conceived and designed the experiments, authored or reviewed drafts of the article, and approved the final draft.
- Huamei Li conceived and designed the experiments, authored or reviewed drafts of the article, and approved the final draft.

## Data Availability

The data is available at figshare and GitHub:

- scAnnoX (2023). scAnnoX的. figshare. Software. https://doi.org/10.6084/m9.figshare.24715233.v3

- https://github.com/XQ-hub/scAnnoX/tree/main/data.

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
