# Peer review of "scAnnoX: an R package integrating multiple public tools for single-cell annotation"

_PeerJ, doi:10.7717/peerj.17184_

## Round 0.1 · original submission · Major Revisions

I suggest authors to go through the comments from all the reviewers and address them in the revised version of the manuscript.

**Language Note:** The review process has identified that the English language must be improved. PeerJ can provide language editing services - please contact us at copyediting@peerj.com for pricing (be sure to provide your manuscript number and title). Alternatively, you should make your own arrangements to improve the language quality and provide details in your response letter. – PeerJ Staff

·

Basic reporting

A satisfactory level of professional English has been used throughout the article.
inclusion of literature references and provide sufficient field background/context have been provided. The article encompasses an ample introduction and background, illustrating how the work aligns with the broader field of knowledge. References relevant to the prior literature have been provided but are insufficient to justify the problem in the field(s) of application. Therefore, the authors are suggested to revise and include all relevant references to strengthen the problem statement. Figures and tables are well presented. Authors are suggested to revise the sequence of figures again carefully so that all figures must be presented Raw data should be shared. The article's structure is well aligned with an acceptable format of 'standard sections,' as outlined in the Instructions for Authors. Figures are pertinent to the article's content, possess sufficient resolution, and are appropriately captioned and labeled. All relevant raw data has been made available by PeerJ’s Data Sharing policy. The article presents relevant results which are about the problem.

Experimental design

The article presents original primary research within the Aims and Scope of the journal.
The research question is well defined well-defined, relevant & meaningful. It is stated how research fills an identified knowledge gap.
The knowledge gap is well identified, and statements made by the authors justify how the study contributes to filling that gap. However, most of the sub-sections in the materials and methods section were not supported with the relevant citations.
Rigorous investigation performed to a high technical standard. The investigation has been conducted rigorously and to a high technical standard.
Methods described with sufficient detail & information to replicate.
Methods have been described with sufficient information to be reproducible by another investigator. Appropriate citations must be included to strengthen the experimental section.

Validity of the findings

Ten extensively utilized algorithms have been developed to identify cell identities within single-cell RNA sequencing data. The results are quite novel and rational. All underlying data have been provided and the generated R package can be accessed through the links provided by the authors. The data (or raw data) and package presented are robust, statistically sound, & controlled. Conclusions are well described and are linked to the objective.

Additional comments

The main review findings are summarized as follows:
1. The introduction references are insufficient to justify the problem in the field(s) of application.
2. Sub-sections under the material and methods section lack appropriate citations, making it unclear whether the current method of data processing was designed by authors for the first time or reported previously.
3. The discussion section requires improvement in its clarity and coherence. Please elaborate on your results, drawing comparisons to justify the problem you addressed. Bridge the gaps identified in previous reports within the fields of cell or molecular biology.
Please advocate your results by citing appropriate reports or developments previously published. Or justify the limitations observed in previous sequencing technologies in omics (RNA sequencing/Transcriptomics etc).

Reviewer 2 ·

Basic reporting

The language in the main text is acceptable. However, abstract needs to be re-written. It does not sound objective.The authors should use statements like "meticulously" or "greatly facilitates". These sound over-pushed and subjective. Apart from this, the language/grammar in the abstract is not acceptable.

Experimental design

The paper brings many single cells tools together and brings a blended multiversion. It has been designed appropriately around this objective.

Validity of the findings

Findings are acceptable.

·

Basic reporting

Check additional comments

Experimental design

Check additional comments

Validity of the findings

Check additional comments

Additional comments

In this work, the authors aim to propose an R package “scAnnoX”, which combines the results of 10 different algorithms for single-cell data analysis. By applying the R package to different datasets, the authors have shown the superiority of the package. Reviewer feels that the manuscript may have been well executed at the lab level, however, the authors have not conveyed the message that they intended to. My comments need to be addressed by the authors before considering the manuscript for publication.

1) The R package on GitHub cannot be installed, I tried with two different PCs (windows), please fix it.
2) The introduction section can be improved, in the second paragraph, the authors mentioned “One approach involves …”, “The second method …”, “A recent and noteworthy approach …”. It’s confusing since it’s not clear if the authors intended to introduce different types of methods or different specific methods. If different types of methods, please make it clear, and include several references for each type. If the authors indicated specific methods, please provide the name of the method, and include the corresponding reference.
3) Line 121, “Each of these algorithms has unique strengths and applications”, could the authors elaborate a little bit regarding this, and could consider moving these to the introduction section.
4) Line 128, “After the aggregation, the frequency …”, I am confused here, is this like a voting algorithm? Please make it clear.
5) Line 149, “Diverse performance metrics were employed …”, please list the performance metrics used here.
6) Line 163, RMSE is also a performance metric, but it’s for continuous rather than categorical variables. It’s not appropriate to use RMSE here.
7) Line 179, “To facilitate users in selecting the most appropriate algorithm …”. Based on my understanding, this R package combines the results, not select the optimal algorithm. Please edit the language here.
8) Line 187, for Figure 1, the authors should add more details to explain how it works.
9) English should be improved, and please check the grammar errors throughout the manuscript, and please remove the “-” from words like “pro-vided” (line 278) and “be-tween” (line 305).
10) Figure 2A-D, it’s better to list the corresponding accuracy on the top of each histogram.
11) Figure 3A, is it based on the predicted results? If so, please revise the language and make it clear. And I suggest the authors highlight the real label and put it aside.

---

## Round 0.2 · accepted · Accept

The manuscript is acceptable at this stage.

·

Basic reporting

It is highly appreciated that authors have done a through revisions and professional English has been used throughout the article now. Sufficient field background/context has been provided. The article encompasses an ample introduction and background, illustrating how the work aligns with the broader field of knowledge.

References relevant to the prior literature have been provided by authors to justify the problem in the field(s) of application in response to my first review comments. I highly appreciate their commitment to scientific spirit.

Experimental design

Already found quite sound during first review. However, references were missing which have been provided by the authors. After inclusion of the relevant citations for the methods used, this section is now quite complete.

Validity of the findings

Ten extensively utilized algorithms have been developed to identify cell identities within single-cell RNA sequencing data. The results are quite novel and rational. All underlying data have been provided and the generated R package can be accessed through the links provided by the authors. The data (or raw data) and package presented are robust, statistically sound, & controlled. Conclusions are well described and are linked to the objective.
The R package developed by authors is downloadable which is highly appreciated.

Additional comments

Not additional comments required. Discussion has been improved well. The annotation tools contained within scAnnoX are currently limited so citations are not mandatory.

Reviewer 2 ·

Basic reporting

The manuscript is now suitable for publication.

Experimental design

The manuscript is now suitable for publication.

Validity of the findings

The manuscript is now suitable for publication.

Additional comments

The manuscript is now suitable for publication.

·

Basic reporting

The authors have addressed my comments.

Experimental design

The authors have addressed my comments.

Validity of the findings

The authors have addressed my comments.